# Advancing EDGE Zones to identify spatial conservation priorities of tetrapod evolutionary history

Sebastian Pipins [1,2,3,4] ✉, Jonathan E. M. Baillie[1], Alex Bowmer[1,5], Laura J. Pollock [6,7], Nisha Owen[1,9] & Rikki Gumbs [3,8,9]

The biodiversity crisis is pruning the Tree of Life in a way that threatens billions of years of evolutionary history and there is a need to understand where the greatest losses are predicted to occur. We therefore present threatened evolutionary history mapped for all tetrapod groups and describe patterns of Evolutionarily Distinct and Globally Endangered (EDGE) species. Using a complementarity procedure with uncertainty incorporated for 33,628 species, we identify 25 priority tetrapod EDGE Zones, which are insufficiently protected and disproportionately exposed to high human pressure. Tetrapod EDGE Zones are spread over five continents, 33 countries, and 117 ecoregions. Together, they occupy 0.723% of the world's surface but harbour one-third of the world's threatened evolutionary history and EDGE tetrapod species, half of which is endemic. These EDGE Zones highlight areas of immediate concern for researchers, practitioners, policymakers, and communicators looking to safeguard the tetrapod Tree of Life.

With over half of Earth's land surface exposed to high levels of human pressure[1] and the current protected area network not sufficient to avert the present extinction crisis[2,3], determining which areas should be prioritised for conservation is key. Typically, prioritisations have reflected patterns of endemism, extinction risk, and species richness[4,5], and tend to overlook spatial patterns of evolutionary history, an important[6–8] and largely neglected[9–11] component of biodiversity. Evolutionary history is being disproportionately lost in a way that is threatening deep evolutionary branches across the Tree of Life[12,13]. This entails not only a loss in our natural heritage but also a loss in the diversity of biological features and functions resulting from divergent evolutionary histories[14], many of which sustain humanity's wellbeing[15–17]. Maintaining evolutionary history also preserves living variation, providing 'option value' in the form of unexplored or potential benefits for future generations[18–20]. It is therefore essential to identify and conserve areas harbouring large accumulations of threatened evolutionary history.

The dynamics of evolutionary history have been mapped in several ways, based on the measure Phylogenetic Diversity (hereafter, PD) which sums the lengths of phylogenetic branches to quantify the unique and shared evolutionary history of species[18]. While spatial patterns of PD are intrinsically correlated with species richness[21–23], the two aspects of biodiversity can decouple locally[15,24] and highlight non-congruent priority areas[25,26]. Hotspots of range-restricted PD, referred to as 'phylogenetic endemism'[27], display low levels of protection[28] and tend to occur in regions of high human pressure[28,29]. This pattern of low protection is surprisingly common amongst important areas of evolutionary history[26,30] and, in the Americas, the protected area network captures less terrestrial vertebrate PD than expected by chance[31]. In contrast, Biodiversity Hotspots[4] may serve as better surrogates for phylogenetic diversity[31,32]). Biodiversity Hotspots were not designed with PD explicitly in mind, however, and no quantification of their varying

[1]On the Edge, London, UK. [2]Royal Botanic Gardens, Kew, London, UK. [3]Department of Life Sciences, Imperial College London, Ascot, Berkshire, UK. [4]Science and Solutions for a Changing Planet DTP, Grantham Institute, Imperial College London, London, UK. [5]Department of Global Health & Development, London School of Hygiene and Tropical Medicine, London, UK. [6]Department of Biology, McGill University, Montreal, Quebec, Canada. [7]Quebec Centre for Biodiversity Sciences, Montreal, Quebec, Canada. [8]EDGE of Existence Programme, Zoological Society of London, London, UK. [9]These authors contributed equally: Nisha Owen, Rikki Gumbs. ✉e-mail: sebastianpipins@ontheedge.org

evolutionary histories for all tetrapod groups (i.e., amphibians, birds, mammals, and reptiles) has been carried out.

Spatial priorities of evolutionary history can also be revealed by taking a species-specific approach and identifying areas containing high concentrations of the threatened species who contribute disproportionately to the Tree of Life. One way of valuing species in this way is through the EDGE approach, which ranks species based on a combination of their Evolutionary Distinctiveness (ED) and their Global Endangerment (GE)[33]. These ranked lists exist for a variety of taxonomic groups, including mammals[33,34], amphibians[35], birds[36], corals[37], reptiles[38], gymnosperms[39], and sharks and rays[40]. The approach has recently been updated under the EDGE2 methodology[41] to incorporate phylogenetic complementarity[41], which describes how the irreplaceability of focal species is influenced by the extinction risk of closely-related species[42,43]. Species with many secure close relatives are considered less irreplaceable than those with few and highly threatened relatives. As such, EDGE2 scores (hereafter, referred to as EDGE scores) now quantify threatened evolutionary history, with the scores representing the amount of PD expected to be lost, in millions of years (MY), that can be averted with conservation action[41]. The approach also allows for the incorporation of species with inadequate data: those lacking phylogenetic information are imputed across a distribution of trees, and those missing extinction risk data receive estimated probabilities of extinction with a median value comparable to the Vulnerable IUCN categorisation. All incorporated species therefore receive an EDGE score but only those that meet the following two criteria are considered EDGE species[41]: (1) they are in an IUCN threatened category (Vulnerable, Endangered, Critically Endangered) or are classified Extinct in the Wild; and (2) they have an EDGE score above the median for all species in their clade with >95% certainty.

There have been several studies mapping the spatial distributions of EDGE species[10,23,30,44], although few consider a wide scope of taxonomic groups (but see ref. 30). Previously, EDGE species have been used to identify priority regions of mammal and amphibian evolutionary history, termed 'EDGE Zones', by Safi et al.[23]. In our study, we also produced a set of priority sites using a refined methodology that accounts for uncertainty in the data and that now incorporates all tetrapod groups. Our method prioritised for large accumulations of unique threatened evolutionary history, whether driven by a small number of highly distinct species or a larger number of less distinctive but more threatened species as both represent important potential losses from the Tree of Life. We consider our approach an update to and extension of Safi et al.'s prioritisation, with the same principal aim of conserving evolutionary history. As such, we retain the term 'EDGE Zones' for the priority regions identified.

In this study, we also explore patterns of EDGE species richness for all tetrapod groups globally, assessing patterns of endemism and ranking nations by their pooled EDGE richness. We then describe patterns of threatened evolutionary history globally and within Biodiversity Hotspots, revealing areas harbouring disproportionate levels of expected loss. Within each EDGE Zone, we quantify the patterns of threatened PD, levels of protection, and extent of human pressure. In formulating these areas, our intention is to highlight global priorities for threatened evolutionary history as an important component of conservation prioritisation, especially in those EDGE Zones currently overlooked by existing conservation efforts, in order to safeguard the Tree of Life.

## Results
### EDGE species richness
We mapped the distributions of 2937 EDGE tetrapod species (919 amphibians, 683 birds, 618 mammals, and 717 reptiles), representing 98.2% of tetrapods that meet the EDGE species criteria under the EDGE2 protocol (2992 spp. total)[41,45]. Their collective distribution covers 92.9% of the world's terrestrial surface (Fig. 1a) and 833

ecoregions (median: 15 species per ecoregion; IQR: 8–27 species; Supplementary Fig. 1). EDGE species richness increases towards the equator (Supplementary Fig. 2) and is particularly high across large parts of Southeast Asia and the Indo-Gangetic plain, as well as in Hispaniola, the highlands of Cameroon, and the Eastern Arc mountains of East Africa (Fig. 1a). Maximum EDGE species richness occurs in Northern Madagascar, with a single 96.5 km × 96.5 km grid cell containing 45 EDGE species, but EDGE species are absent from parts of Central Australia, Central Europe, and the Sahara Desert. EDGE species richness peaked in the Madagascar Subhumid Forest ecoregion with 196 species (Supplementary Fig. 1).

Madagascar (317 spp.), Mexico (241), and Indonesia (191) contained the highest number of EDGE species (Fig. 1b; see "Data availability"), and 75.6% of EDGE species are endemic to a single country (2262 of 2992). However, the pattern varies by taxonomic group; 90.1% of EDGE amphibians are country endemics, followed by 82.4% of EDGE reptiles, 65.5% of EDGE mammals, and 58.5% of EDGE birds. EDGE species endemism is also notable at a grid cell level, with the distribution of 1451 EDGE species (49.4% of total) limited to any one 96.5 km × 96.5 km grid cell. The patterns also vary for each individual taxonomic group (Supplementary Fig. 3), although certain areas show co-occurrence for each group (Supplementary Fig. 4), including Mesoamerica, the Caribbean, the Andes, the Guinean Forests, the Eastern Arc, Madagascar, the Western Ghats, Sri Lanka, and Southeast Asia.

### Threatened evolutionary history
To estimate the amount of threatened evolutionary history that could be secured with conservation action we mapped the summed EDGE scores of 33,628 tetrapod species, representing 92.4% of all 36,376 tetrapod species with EDGE data (Fig. 2a; Supplementary Fig. 5). Several approaches to mapping threatened evolutionary history have involved summing branch lengths from a phylogenetic tree[25,27–29,46] though other studies have summed species-specific values[23,30,47]. We found a very strong positive correlation between both the phylogenetic branch length approach and the summed EDGE score approach for calculating threatened evolutionary history for each tetrapod group (Pearson's correlation adjusted for spatial autocorrelation[29,48–50]; all $\rho > 0.99$) (Supplementary Note 1), as well as a high overlap in the location of priority grid cells at the 90th (>94.8% overlap) and 95th percentile (>94.4% overlap) for each taxonomic group (Supplementary Note 1).

As with EDGE species richness, threatened evolutionary history increases towards the equator (Supplementary Fig. 2). Southeast Asia, Africa and the Indian subcontinent have a higher incidence of extinction risk relative to the evolutionary history present compared with areas such as South America, where the proportion of evolutionary history that is threatened is lower (Fig. 2b). The top-scoring grid cell is in Cameroon, with 703 MY of threatened evolutionary history. Other areas of importance include Honduras, Hispaniola, the Atlantic Forest, the Western Ghats of India, and Sri Lanka.

The distribution of threatened evolutionary history was highly correlated with species richness (Pearson's correlation: r = 0.916, e.d.f. = 36.3, $p < 0.0001$). However, priority regions resulting from the two measures are more dissimilar than the high correlation suggests, with the congruence decreasing at high percentiles. For example, for the 80th percentile (i.e., the 20% of highest scoring grid cells), the two have an 84.8% overlap. However, at the 90th percentile this decreases to an overlap of 69.8%, followed by 49.4% at the 97.5th percentile. Areas of high species richness dominate large parts of the Amazon basin and the Atlantic Forest, reflecting widely distributed shared species compositions (Supplementary Fig. 6). Meanwhile, areas of high threatened evolutionary history at the 97.5th percentile show a wider array of geographic locations, including Southeast Asia, Cameroon, Madagascar, the Eastern Arc, the Western Ghats, and Sri Lanka

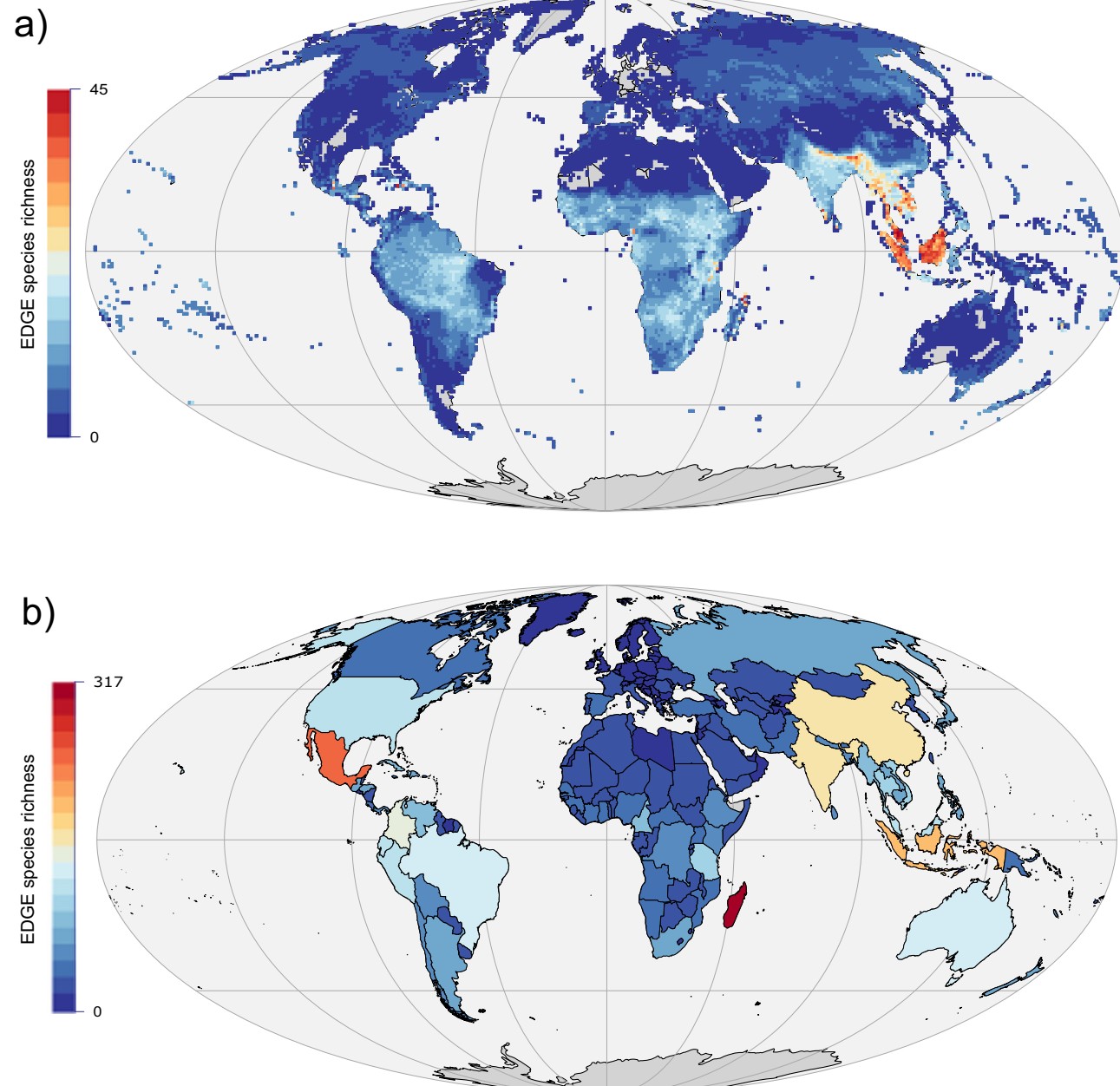

**Fig. 1 | The species richness of Evolutionarily Distinct and Globally Endangered (EDGE) tetrapods.** The richness of 2937 tetrapod EDGE species mapped (**a**) using a 96.5 km × 96.5 km equal area grid and (**b**) at the national level in a Mollweide projection. Source data are provided as a Source data file.

(Supplementary Fig. 6). This dissimilarity in high scoring areas underlines the importance of considering multiple facets of biodiversity within conservation prioritisations[30,51,52].

**Tetrapod EDGE Zones**

We used spatial complementarity to identify grid cells containing highly irreplaceable and threatened evolutionary history (see "Methods"). This highlighted 32 priority grid cells, whose constituent species together represented 25% of total threatened tetrapod evolutionary history (our chosen threshold). The complementarity procedure was then repeated 1000 times using a distribution of EDGE scores to account for the uncertainty in phylogenetic, extinction risk, and spatial data. We found that the irreplaceability (frequency of selection, measured between 0 and 1) of the initial 32 priority cells had a median value of 0.79 (Supplementary Fig. 7); four grid cells, found in Northern Madagascar, Hispaniola, Mexico, and Sri Lanka, were selected in every

one of the 1000 iterations and a further nine grid cells were selected in more than 90% of iterations. The uncertainty analysis altogether selected from a pool of 320 grid cells, ranging from an irreplaceability of 0.001 to 1 and with a median of 0.007 (Supplementary Fig. 7).

Geographically proximate clusters of priority cells were then grouped and joined with contiguous cells (those neighbouring from any direction) from the uncertainty analysis (Supplementary Fig. 7). Eight clusters, in Sri Lanka, Seychelles, Southern Madagascar, Central Madagascar, Ethiopia, Colombia, Mexico, and the Atlantic Forest of Brazil, were unaffected by this and did not increase in size. The remaining 17 clusters were extended by the inclusion of contiguous cells, by up to a maximum of 14 additional cells in the case of the Gangetic Plains. Across the priority clusters, we found that both the proportion of endemic species and the summed EDGE score were significant predictors of the irreplaceability values of the grid cells within (Supplementary Note 2).

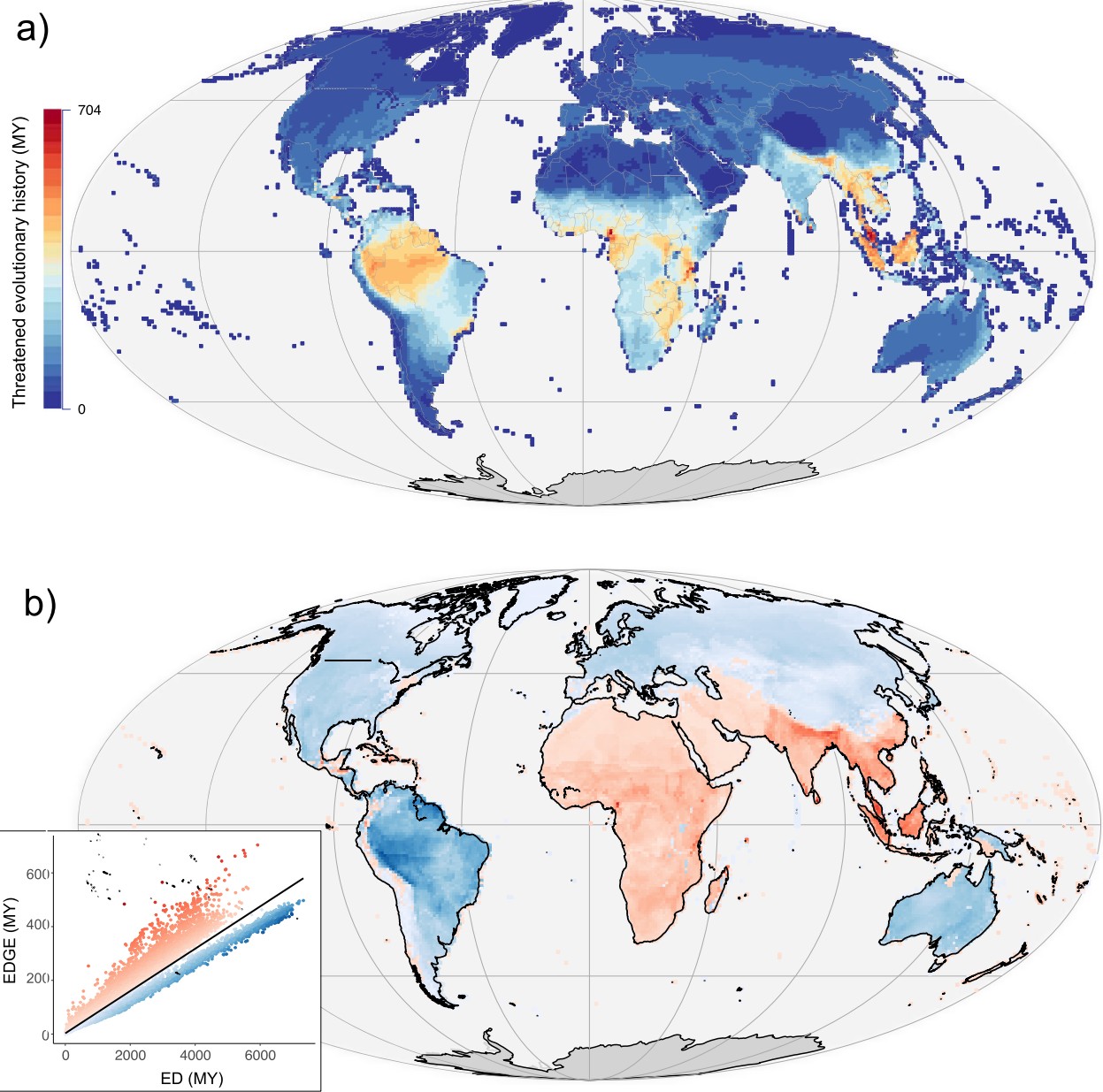

**Fig. 2 | The global distribution of threatened tetrapod evolutionary history.**
**a** Threatened tetrapod evolutionary history mapped using summed Evolutionarily Distinct and Globally Endangered (EDGE) scores[41]; **b** a linear regression of global patterns of summed EDGE scores against summed Evolutionary Distinctiveness (ED) scores[41], where warm coloured residuals reflect areas where evolutionary history is more threatened than expected given the summed ED of the region, and cool coloured residuals reflect where evolutionary history is less threatened than expected given the summed ED. Source data are provided as a Source data file.

Overall, our procedure resulted in 25 'EDGE Zones' comprising 112 grid cells (Fig. 3). These EDGE Zones are spread across five continents and 33 countries, covering 0.723% of the Earth's terrestrial surface. They are located over 117 ecoregions and 10 different biomes, with 23 EDGE Zones overlapping with the Tropical & Subtropical Moist Broadleaf Forests biome, eleven overlapping with Mangroves, and eight overlapping with Tropical & Subtropical Dry Broadleaf Forest. Of the 109 developing countries with global Multidimensional Poverty Index data[53], which assesses a person's combined deprivation in health, living standards, and education, 29 of these overlap with 24 EDGE Zones (all except the Australian EDGE Zone). The score for these countries falls between the 6th and 94th percentile, with the median being the 42nd percentile.

The priority cells selected in our EDGE Zones analysis (referred to here as EDGE Zone priority cells) were robust to resolution size, mode of calculation, and metric choice: 83% of grid cells selected using a coarser resolution of 193 km × 193 km were overlapping or contiguous to EDGE Zone priority cells (Supplementary Fig. 8); there was a 97.7% overlap between EDGE Zone priority cells and cells selected using phylogenetic branch-length calculations of expected PD loss (Supplementary Fig. 9), with the two methods showing a strong correlation in the frequency in which cells were selected ($\rho = 0.958$, $p < 0.0001$); there was a 68.8% overlap between EDGE Zone priority cells and cells selected using EDGE scores weighted by range size ('EDGE rarity'; Supplementary Fig. 10). When the complementarity procedure was repeated but instead selected for weighted endemism, we found that it

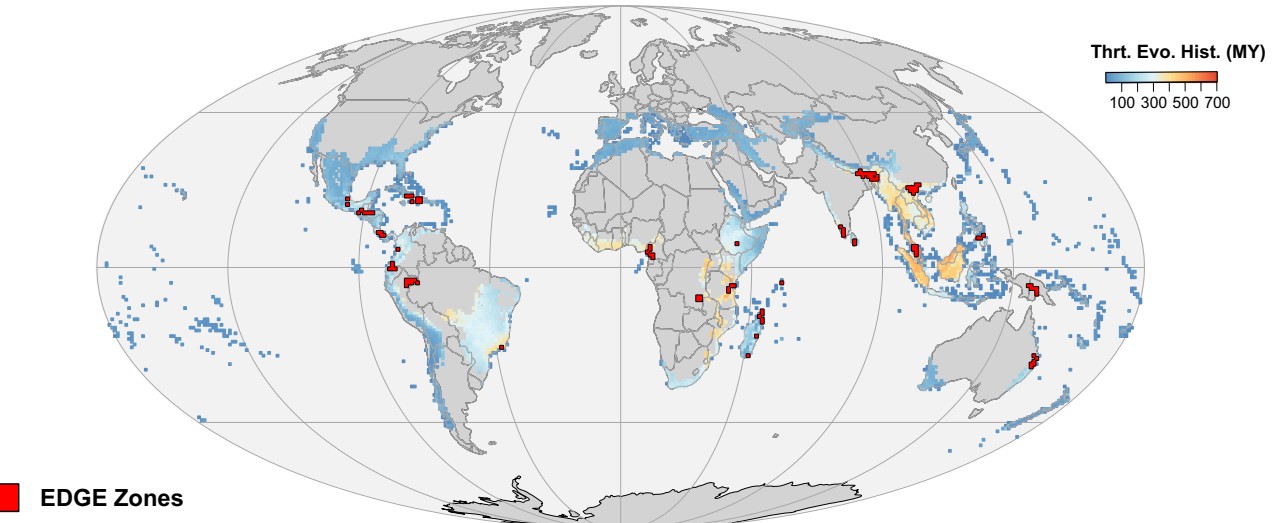

**EDGE Zones**

**Fig. 3 | Tetrapod EDGE Zones.** The locations of tetrapod EDGE Zones, a complementary set of 25 grid cell clusters containing large and unique accumulations of tetrapod threatened evolutionary history. Here, tetrapod EDGE Zones are superimposed over existing Biodiversity Hotspots, where colours represent per grid cell threatened evolutionary history, increasing from shades of dark blue through to orange. Source data are provided as a Source data file.

took 37 priority cells to represent 25% of species richness (Supplementary Fig. 11), with >50% of these (19/37) not shared with the EDGE Zone priority grid cells. The 37 cells identified contained 9522 MY of threatened evolutionary history (23.3% of the total).

In total, there are 11,662 tetrapod species found in EDGE Zones (34.7% of total), 1717 threatened species (26.8%), and 918 EDGE species (31.3%) (Supplementary Data 1; see "Data availability"). In every case, this is more than the random expectation; EDGE Zones capture 40% more species, 437% more threatened species, and 348% more EDGE species than would be expected were EDGE Zones distributed at random. The Zone with the most EDGE species is Northern Madagascar with 122 (Fig. 4a), 29.8% more than the next highest scoring Zone in Hispaniola with 94.

Together, if all the species that have at least a part of their range in an EDGE Zone were saved, this would secure 33.3% of total threatened tetrapod evolutionary history (equivalent to 16,820 MY, and 265% more than the 6345 MY if EDGE Zones were drawn at random). This includes 37.2% of threatened bird evolutionary history, along with 34.3% for amphibians, 31.6% for reptiles, and 31.4% for mammals. Again, Northern Madagascar is the EDGE Zone with the largest accumulation of threatened evolutionary history with 1339 MY, followed by Guatemala-Honduras with 1188 MY (Fig. 4b).

Naturally, the size of EDGE Zones mean they do not capture the full ranges of all constituent species; for those non-endemic species found within EDGE Zones, the proportion of their ranges contained within the 25 EDGE Zones is a median of 13.5% for amphibians, 6.96% for reptiles, 4.19% for mammals, and 3.57% for birds. Collectively, EDGE Zones contain a median of 4.73% of the ranges of the non-endemic tetrapod species found within. However, endemism levels are notable; EDGE Zones harbour 1457 grid cell endemics, which is 2749% more than the random expectation of 53 endemic species. Furthermore, 1727 species are endemic to anywhere within an EDGE Zone (14% of all species found within an EDGE Zone), meaning 15.5% of total threatened tetrapod evolutionary history is contained exclusively within EDGE Zones. The most endemics are found in Costa Rica-Panama (186), Northern Madagascar (185), and Sri Lanka (180) (Fig. 4a). Furthermore, 52.2% of EDGE species in EDGE Zones are endemic to a single Zone (480 of 918 species), equating to 16.0% of all EDGE tetrapods (480 of 2992).

Using the Human Footprint Index[54], we then explored the proportion of EDGE Zones experiencing appreciable levels of human disturbance (scores of above 4 or more, see methods) (Fig. 4c). We found that 81.5% of the total area covered by EDGE Zones is under high human pressure and that this varies by EDGE Zone; from a low of 16.2% to a high of 100%, with the median level of high human pressure in EDGE Zones being 85.3% (Supplementary Data 1). Three EDGE Zones (Sri Lanka, Cuba, and the Western Ghats) are entirely under high human pressure. Between 1993 and 2009, a mean shift of 7.23% of the land within EDGE Zones transitioned from low to high human pressure; but this also varied, with 5.8% of the land within Peninsular Malaysia shifting to lower human pressure, compared to 32.8% of the area within New Guinea shifting to high human pressure. The human pressure in EDGE Zones is higher than the background expectation of 56.6%.

A mean of 20% of EDGE Zone land is under any form of protection (relaxed protection), while 10% is under stricter protection standards (IUCN I:IV) (Fig. 4c). When assessing the protection standards within each EDGE Zone grid cell, we found levels of relaxed protection were comparable to the United Nations Convention on Biological Diversity's (CBD) Aichi target of 17% ($t(111) = -0.232$, $p = 0.817$)[55], but were significantly below the target when based on stricter protection standards ($t(111) = -6.73$, $p < 0.0001$). Both relaxed and stricter protection levels were significantly below the CBD's Kunming-Montreal 2030 milestone of 30% ($t(111) = -8.11$, $p < 0.0001$; $t(111) = -16.5$, $p < 0.0001$)[56]. DR Congo saw the highest cover of any form with protection at 53%; New Guinea and Colombia have less than 1% of their area protected, and four Zones (New Guinea, Colombia, the Western Ghats, and Ecuador) show zero protection when stricter protection categories were considered. However, in some cases these scores are an artefact of inconsistencies in national data reporting, given that India withholds data on 900 protected areas, influencing the low score seen in the Western Ghats (https://www.protectedplanet.net/country/IND). Even within the protected portions of EDGE Zones, high human pressure levels are found across 75.7% of their extent, rising to 83% for the non-protected portions.

Within EDGE Zones, amphibians have significantly greater median EDGE scores per grid cell compared with the other tetrapod classes ($p < 0.001$ for all pairwise comparisons made using ANOVA with Tukey's Honest Significant Difference Test; Supplementary Note 3), but outside of EDGE Zones, the median EDGE scores of reptiles are significantly greater ($p < 0.0001$). Relative to their richness within EDGE Zone grid cells, however, there was no difference between the

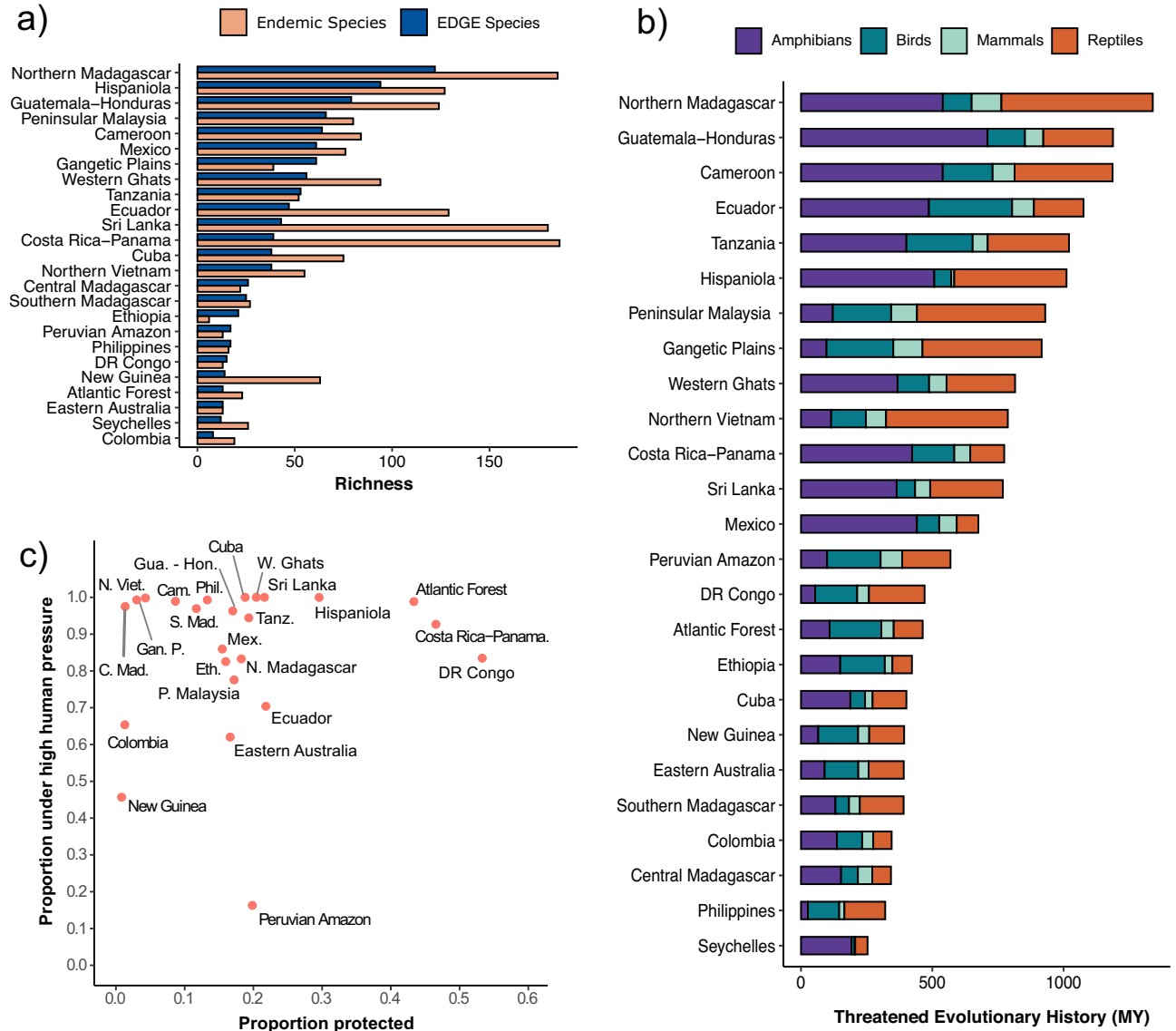

**Fig. 4 | The diversity, human pressure, and threatened evolutionary history within tetrapod EDGE Zones. a** The richness of EDGE species and endemics species within EDGE Zones. **b** The threatened evolutionary history (in millions of years; MY) across different EDGE Zones, calculated using the combined EDGE scores of all species found within each Zone and displayed in terms of the different tetrapod groups; **c** the proportion of each EDGE Zone that is under any form of protection and that is experiencing high levels of human pressure. The Seychelles EDGE Zone was excluded from (**c**) due to absent human footprint data in this location. Source data are provided as a Source data file.

amphibian and reptile proportional contribution to threatened evolutionary history ($p = 0.998$), although both contributed significantly more than mammals and birds ($p < 0.0001$; Fig. 5; Supplementary Note 3).

### Biodiversity Hotspots

Biodiversity Hotspots, as defined by Myers et al.[4], cover 18.5% of the world's terrestrial surface, capturing 73.5% of EDGE tetrapods and 74.5% of tetrapod threatened evolutionary history (Supplementary Fig. 12). The distribution of this diversity is unevenly split across the 36 different Hotspots (Supplementary Data 2). For instance, the three Hotspots with the highest median threatened evolutionary history (relating to the median EDGE scores of grid cells found within each hotspot) are Sundaland (393 MY), Indo-Burma (354 MY), and the Guinean Forests of West Africa (339 MY). These same three hotspots scored highest for peak threatened evolutionary history (the maximum summed EDGE grid cell found in each hotspot), with the Guinean

Forests of West Africa ranked first (Fig. 6). In terms of EDGE tetrapod richness, Madagascar and the Indian Ocean Islands (334 spp.) ranked first, followed by the Caribbean (300 spp.), and Mesoamerica (297 spp.). All three metrics considered were positively correlated with species richness (maximum summed EDGE – $r(34) = 0.6$, $p = 0.0001$; median summed EDGE – $r(34) = 0.45$, $p = 0.0001$; EDGE tetrapod species richness – $r(34) = 0.64$, $p < 0.0001$).

## Discussion

In this study, we mapped EDGE species from all tetrapod groups, revealing a high endemicity at both a national and grid cell level. Spatial patterns of threatened evolutionary history diverge from species richness at the highest-ranking priority grid cells, with key areas for threatened evolutionary history including Cameroon, Tanzania, and Hispaniola. To drive the conservation of threatened evolutionary history, we identified 25 priority regions of disparate communities of species, termed tetrapod 'EDGE Zones', which together harbour 33.3%

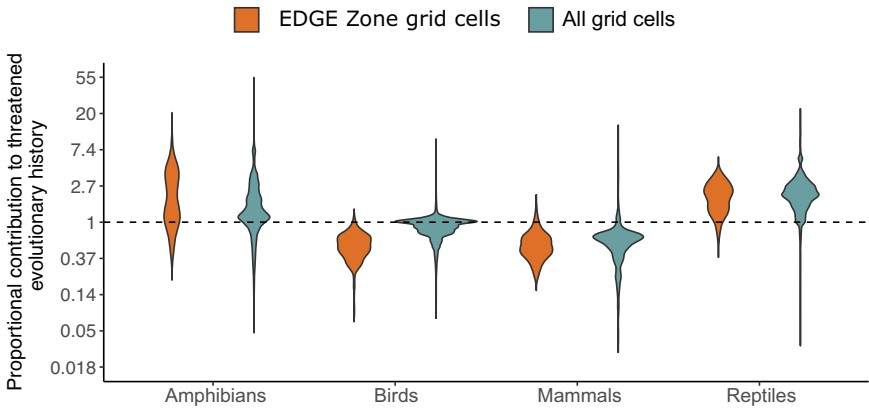

**Fig. 5 | The proportional contribution of tetrapods to the threatened evolutionary history within and outside of EDGE Zone grid cells.** The proportional contribution of each tetrapod group to the threatened evolutionary history of EDGE Zone grid cells and all grid cells, where a score of >1 means that a group contributes more to the threatened evolutionary history of a cell than expected from its per cell richness: for example, if amphibians represent 30% of a cell's species richness but contribute 60% to threatened evolutionary history, their proportional contribution is 2. Source data are provided as a Source data file.

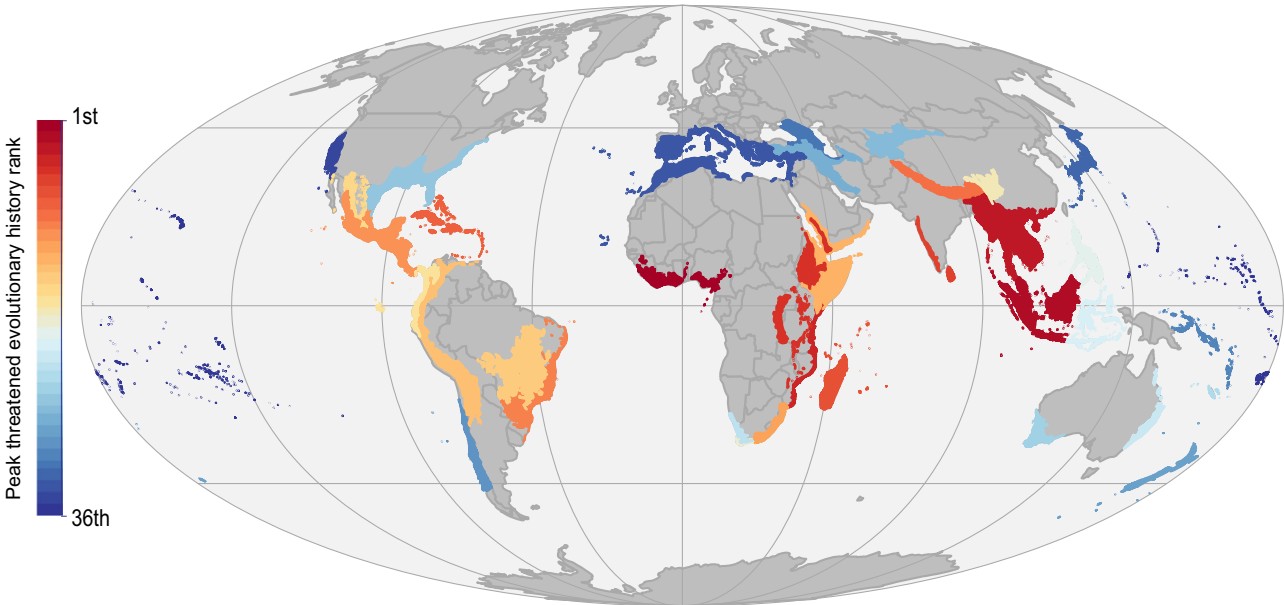

**Fig. 6 | Biodiversity Hotspots ranked by peak threatened evolutionary history.** The rank of all 36 Biodiversity Hotspots, as defined by Myers et al.[4], was measured by comparing the grid cells in each with the largest amounts of threatened evolutionary history and coloured from first in dark red to last in dark blue. Source data are provided as a Source data file.

of threatened evolutionary history and 31.3% of EDGE tetrapods, with approximately half of both being found nowhere else. Concerningly, high levels of human pressure are ubiquitous across these Zones, and protection levels largely fail to meet globally recognised targets for 2020 and 2030[56].

Given Madagascar's highly endemic biodiversity[57], phylogenetic importance[28,30], and high levels of extinction risk[58], it is unsurprising that the island harbours the greatest number of priority EDGE tetrapods (10.4% of total). This includes important EDGE species such as the 2nd top-ranked EDGE reptile, the Madagascar Big-headed Turtle (*Erymnochelys madagascariensis*; EDGE score: 84 MY), and the 2nd top-ranked mammal, the Aye-aye (*Daubentonia madagascariensis*; EDGE score: 20 MY). However, high micro-endemism within Madagascar[59,60] means the EDGE richness here is isolated in patches along the Northern and Eastern parts of the island. In contrast, very large areas of Southeast Asia show elevated levels of EDGE species, reflecting how the

'looming biodiversity disaster' in this region is impacting highly unique and wide-ranging species across extensive parts of their range[61].

Half of EDGE tetrapods are endemic to any one grid cell and three-quarters are found exclusively in a single country. Given that environmental policy is usually set at the national level[62], our findings suggest that nations must show leadership in protecting their most evolutionarily distinct species. By protecting EDGE species locally, and thus their habitat and other sympatric species, nations can not only preserve their unique biodiversity and the benefits provided by it, but they can also safeguard the global values captured by maintaining the Tree of Life[8,17].

We elected to use summed EDGE scores over a branch-based approach to calculate threatened evolutionary history. The two approaches were highly correlated and displayed 98% overlap when selecting EDGE Zone priority grid cells. Summing EDGE scores does mean there will be instances of overcounted branch lengths. However,

due to the way EDGE scores are calculated, this double counting is most acute for Critically Endangered sister species and becomes increasingly inconsequential as related species become less threatened. For example, an internal branch subtended by at least one Least Concern species can only contribute a maximum 6% of its total length[41], as the low extinction risk of the Least Concern species effectively safeguards its descendant branches against extinction. Summing EDGE scores can therefore slightly overestimate the amount of expected loss in areas with large concentrations of highly threatened, closely-related species. Given the large potential losses to deep evolutionary history that this could entail[63,64], we are not concerned by this methodological artefact. Furthermore, we believe the summed EDGE2 approach provides a clear benefit to policymakers and conservation practitioners. This approach, which uses publicly available data[45], is more accessible to non-specialists both in theory and application and can be computed at a fraction of the time of branch-length-dependent methods, providing a feasible avenue to incorporate evolutionary history more readily within conservation planning and decision making. Considering species are the typical unit of conservation, using species-specific scores also allows for the inclusion of other species-level biodiversity measures, such as cultural value and functional traits[52,65], or the incorporation of species from disparate clades into prioritisation exercises[36].

The expansion of our EDGE Zones to incorporate adjacent areas with similar communities of species based on the uncertainty in the phylogenetic and extinction-risk data ensures that these prioritisations are robustly delineated to highlight areas of utmost concern. We can say this with confidence because the prioritisations were robust to the uncertainty in the underlying data, to method choice of calculating threatened evolutionary history, and to grain size used in the complementarity procedure. Using complementarity also ensured that EDGE Zones captured disparate species compositions but unlike an approach that selects just for weighted endemism, EDGE Zones efficiently prioritize areas for their unique evolutionary history as well; we found that only 56.3% of grid cells were shared between our chosen approach and one that focussed on weighted endemism only (Supplementary Fig. 11), with the latter also requiring 15.6% more grid cells to capture 6.8% less threatened evolutionary history. We found that this endemism prioritisation over-selected across the Andes, which displays high species turnover but shows diminishing returns on the amount of new phylogenetic diversity represented. Furthermore, it neglected areas of high richness and low grid-cell endemism (e.g. the Amazon) and areas of low richness and high phylogenetic diversity (e.g. Seychelles). We believe this underpins the strength of prioritising using summed EDGE scores, which highlights areas of large concentrations of unique and threatened evolutionary history, whether primarily driven by threat, richness, endemism, or evolutionary distinctiveness.

Although only a fraction of the ranges of non-endemic constituent species are captured within Zones (-5%), 42% of the threatened species and 52% of the EDGE species found within are endemic. As such, roughly 1/6th of global threatened tetrapod evolutionary history is found exclusively in EDGE Zones, and 1/3rd of threatened tetrapod evolutionary is contained partially within EDGE Zones. EDGE Zones are therefore not intended to be the only sites prioritised for protection, but rather they are key regions where the failure to protect the biodiversity within will have major consequences for the Tree of Life and the benefits it bestows. Further research will be needed to assess the conservation opportunities within each EDGE Zone to find where interventions are possible and most likely to have an impact.

The Cameroon EDGE Zone harbours the grid cell with the greatest accumulation of threatened evolutionary history, in part driven by the dramatic declines of evolutionarily distinct amphibians in the region[66,67]. This includes four Critically Endangered EDGE puddle frogs (*Phrynobatrachus* spp.) and eight highly threatened EDGE egg frogs

(*Leptodactylodon* spp.). In fact, there is considerable overlap between EDGE Zones and high concentrations of threatened amphibians[68]. These areas of overlap include Mesoamerica, the northern Andes, the Atlantic Forest, Cameroon, the Eastern Arc of Tanzania, Madagascar, the Western Ghats, Sri Lanka, and the Philippines. This can be explained by the patterns of high distinctiveness and extinction risk seen in amphibians; relative to their richness, amphibians contribute significantly more to the threatened evolutionary history of EDGE Zones than birds and mammals. The Caribbean, where 84% of amphibians are threatened with extinction[69], is a good example of this association between distinctiveness and imperilment. Here, the selection of the Hispaniola EDGE Zone was driven by amphibians (Fig. 4). This includes the influence of the highly speciose (202 spp.), threatened (137 spp.), but also evolutionarily distinct (median ED = 12 MY) clade of robber frogs (*Eleutherodactylus* spp.), of which there are 42 EDGE species in this one Zone.

EDGE Zones also coincide with other priority areas, with 22 overlapping with Biodiversity Hotspots. Several Biodiversity Hotspots contain multiple EDGE Zones, reflecting the presence of disparate assemblages of threatened evolutionary history within them; this includes Hotspots in Sundaland, Mesoamerica, the Caribbean, Madagascar and the Indian Ocean Islands. In our comparison of Biodiversity Hotspots, we found that some regions consistently ranked highly in their levels of threatened evolutionary history and EDGE richness. For instance, the Indo-Burma, Sundaland, Eastern Afromontane, Guinean Forests of West Africa, and the Western Ghats and Sri Lanka Biodiversity Hotspots all scored in the top ten for each aspect of evolutionary history we explored. Concerningly, all five of these Hotspots are also ranked in the top third most densely populated Biodiversity Hotspots (>114 people per square kilometre)[70], each has a child malnutrition rate of more than 20%[70], and mounting agroeconomic pressures threaten what little intact vegetation is left in most of these regions[71,72]. This necessitates urgent conservation action, and we believe the comparison provided here can add a useful phylogenetic perspective to this endeavour.

Elsewhere, EDGE Zones coincide with priority locations described in other studies of tetrapod evolutionary history[28,29,47]. All countries forecasted to experience the greatest losses in evolutionary history due to land-use driven species extinctions[47] contain EDGE Zones. Elsewhere we found that 24 EDGE Zones overlap with hotspots of tetrapod phylogenetic endemism[28], and all 25 overlap with priorities of tetrapod human-impacted phylogenetic endemism at the 95th percentile[29]. Thus, although EDGE Zones were not formulated based on patterns of endemism, they also effectively capture range-restricted PD.

Our study echoes previous findings reporting low levels of protection and high human impact across phylogenetically important areas[28–30], with 60% of EDGE Zones being covered by less than 10% of strictly designated protected areas and 76% showing north of 80% of their land under high levels of human pressure (Fig. 4). Furthermore, human populations found within EDGE Zone countries face appreciable deprivation in education, health, and living standards, as measured using the multidimensional poverty index. One particularly concerning case study surrounds New Guinea, a region long recognised for its high biodiversity and relatively intact primary vegetation[73]. We found that one-third of the available land in this Zone saw a shift from low to high human pressure in a 17-year period, driven by expansive deforestation efforts[74] that are projected to cause major losses in the evolutionary history[47]. While the protected area coverage across EDGE Zones currently fails to meet the 2030 target of 30% protection, we did find that relaxed-protection levels were comparable to the Aichi target of 17%. Our research demonstrates that large gains of biodiversity are possible within relatively small additions to the global protected area network. As conservation seeks to protect 30% of Earth's land by 2030, we emphasise that evolutionary history must

be considered; more research is needed to build on this and other work[25,28,47] to identify best path forward for these areas.

The downstream utility of global mapping exercises such as that presented here has recently been called into question due to the lack of clear translation to action on the ground[75]. We believe this is a valid criticism, given that the realised or potential impacts of global mapping research is not commonly reported in the scientific literature. However, we foresee these priority regions guiding future efforts to save the world's most distinctive and imperilled tetrapod species through various downstream uses. First, the EDGE Zones presented here will guide the activities of the charitable organisation On the Edge (www.ontheedge.org), directing their conservation grant-making, regional campaigns, and grantee-led storytelling. Second, EDGE Zones will form part of the decision-making for resource allocation for the Zoological Society of London's EDGE of Existence programme (www.edgeofexistence.org), which has already funded work on over 50 EDGE species found within EDGE Zone countries, with a particular focus on the Gangetic Plains and Cameroon. Third, our method of summing EDGE scores offers the potential for extending this approach to other important taxonomic groups, such as seed plants and fish[39,45,76,77], and therefore allows for the expansion of EDGE Zones into freshwater and marine realms. Furthermore, in these groups, tools such as machine learning algorithms are allowing for comprehensive extinction risk predictions[78], helping to mitigate against the IUCN Red List's limited coverage. Finally, we hope this analysis will inspire in-depth, fine resolution research into the current and future levels of irreplaceability, protection, human pressure, climate change, and conservation potential in each EDGE Zone to catalyse applied conservation action.

The initial formulation of EDGE Zones was conducted under the aim of establishing 'a spatial perspective for an otherwise species-centred conservation initiative'[23] in reference to the EDGE approach of phylogenetically informed species conservation. In revisiting this concept, we have retained Safi et al.'s original aim but significantly extended their approach through a revised prioritisation methodology and the consideration of all tetrapod groups and their threatened evolutionary history. In doing so, we have revealed global patterns in the distribution of highly imperilled PD, highlighting where the greatest potential losses are accumulating, where EDGE species are concentrated, and which Biodiversity Hotspots are especially important in capturing this heritage. In revealing these patterns and prioritisations, we have provided a useful frame of reference for conservationists, policymakers, and scientific communicators seeking to safeguard the Tree of Life.

## Methods

### Species distribution, extinction-risk, and phylogenetic data
We obtained species distribution data from the IUCN Red List of Threatened Species (Version 2021.1) for terrestrial and freshwater mammals and amphibians[79], from BirdLife International for birds[80] (Version 2020.1), and from the Global Assessment on Reptile Distributions for reptiles[29] (Version 1.5). Distribution data were filtered so that only native, breeding, and resident extents of distributions were used, where relevant. Antarctic distributions were excluded. Range extents marked as 'Presence Uncertain' and 'Extinct' were removed. Range polygons were then rasterised into a grid format using a resolution of 96.5 km × 96.5 km with a Mollweide equal area projection. We used EDGE scores from Gumbs et al.[45], generated using the updated EDGE2 approach[41], which sourced phylogenetic data from Jetz and Pyron for amphibians[81], Jetz et al. for brids[36], Upham et al. for mammals[82], Tonini et al. for squamates[83], and Colston et al. for crocodiles and turtles[84]. Species names from the distribution data and phylogenetic data were matched using the taxonomic databases referred to in the EDGE calculation of Gumbs et al.[45]. Our study used only those species with both distribution and EDGE data, resulting in the inclusion of 5614 mammals (89.8% of total), 6809 amphibians

(84.9%), 10,937 birds (99.5%), and 10,268 reptiles (92.4%), together representing approximately 92.4% of tetrapod diversity. All mapping and analyses took place in R version 4.1.0.

### EDGE species richness and threatened evolutionary history
To explore the distributions of EDGE species, we mapped EDGE tetrapods at a grid cell level. EDGE species designations were taken from Gumbs et al.[45,85]. We then explored EDGE species richness at the national level, using national reporting data from the IUCN Red List[79]. We then intersected EDGE species distributions with ecoregions[86], finding that there were 33 EDGE species that did not overlap with any ecoregion; these small-island endemic EDGE species had distributions that were too remote to be covered by the coarse designation of ecoregions. We have made the list of EDGE species found within each ecoregion available (see "Data availability").

EDGE scores are species-specific scores calculated by summing the lengths of branches connecting a species at the tip of the tree to the root in an extinction-risk-weighted phylogeny[41]. As these scores are derived from overall estimations of expected PD loss for the entire clade, it stands that summing these EDGE scores should correlate highly with total expected PD loss of the clade[19]. We therefore ran a series of sensitivity analyses for each tetrapod group to explore how well summing EDGE scores reflects the typical phylogenetic branch length approach for calculating expected PD loss. To calculate expected PD loss, we worked with the extinction-risk adjusted phylogenetic trees used in Gumbs et al.[45]. We worked on a per-grid cell basis, summing the branch lengths from the root node to the tips of the trees subset to only the species found within a given grid cell. This was repeated for a distribution of 1000 phylogenies for each group and the average expected PD loss grid-cell values were then contrasted with the summed EDGE grid-cell values by using a Pearson's correlation adjusted for spatial autocorrelation[29,48–50], and then comparing the overlap in priority grid cells using six percentiles (80th, 85th, 90th, 95th, 97.5th, and 99th).

As this comparison revealed a high congruence and strong correlation (Supplementary Note 1), we proceeded to use summed EDGE scores for our analyses. Given that a species' EDGE score reflects the amount of the PD for which it is responsible (it's Evolutionary Distinctiveness, given by an ED2 score[41], referred to here as ED) that is expected to be lost, we ran a linear regression of EDGE ~ ED to see where there is more threatened evolutionary history than expected given the amount of evolutionary history present in a grid cell[29]. The residuals along this linear regression were mapped, with positive residuals highlighting areas where we are projected to lose more evolutionary history than expected given the modelled relationship.

We also assessed the ED of species with EDGE data but that were missing distribution data. For these species, ED scores were generally skewed towards mid to lower percentiles for each tetrapod group (Supplementary Fig. 13), with the median ED for these data-deficient species ranging from the 37th percentile in birds to the 44th percentile in reptiles. We therefore predict that the data gaps within tetrapod species distributions will not significantly alter the results presented here.

We then mapped all 36 Biodiversity Hotspots[4,5,87] at a 96.5 km × 96.5 km resolution and compared them in terms of their maximum EDGE score, median EDGE score, number of species, number of EDGE species, and number of genera.

### Tetrapod EDGE Zones
To identify priority locations of unique threatened evolutionary history, we iteratively selected top-ranking grid cells with the highest summed EDGE scores using spatial complementarity. With each iteration, the species found within the top-ranked cell from the previous selection were removed from the underlying dataset and the process repeated until a threshold was passed. Our threshold for how

many sites to select in this complementarity procedure was the minimum number whose pooled species composition together represented 25% of tetrapod threatened evolutionary history. Here, threatened evolutionary history was measured using the extinction risk-weighted phylogenetic trees from Gumbs et al.[45]. The threshold was met when the subtree connecting all constituent species found within N priority grid cells had a combined expected PD loss equal to 25% of the total for all tetrapod species. Constituent species were classified as those whose distribution overlapped with a priority cell.

A complementary set of 32 priority cells was selected from this procedure, out of a total set of 3001 cells to capture 100% of tetrapod evolutionary history (Supplementary Fig. 14). Contiguous cells (those touching each other in any direction) were grouped together to form single EDGE Zones. Two adjacent pairs of disjunct cells in Haiti and two adjacent pairs of disjunct cells in Mexico were grouped together, respectively, leaving 25 priority clusters. The choice to group these cells was made because we believe their geographical proximity would make it impractical to treat these regions separately from a conservation perspective.

To determine how robust the selection of our priority cells was to changes in resolution size, mode of calculation, and metric choice, we repeated our complementarity procedure using (1) a coarser resolution of 193 km × 193 km, (2) the phylogenetic branch length-based calculation of expected PD loss, and (3) 'EDGE rarity', relating to a species' EDGE score divided by the number of grid cells with which its range overlaps. We then compared our approach to one that selects for weighted endemism only to determine how different the areas highlighted would be from our summed EDGE score approach. To do this, we used complementarity to iteratively select top-scoring cells until 25% of species richness ($n = 8407$) was represented. A species was considered represented if it had any proportion of its range overlapping with a top-scoring cell.

The complementarity procedure was repeated 1000 times using the distribution of EDGE scores to account for uncertainty. The irreplaceability (the frequency of selection) of the 32 priority grid cells was recorded, giving an indication of how confident we are that a site contains a significant amount of threatened evolutionary history not found elsewhere. Proximate priority grid cells were grouped with contiguous cells from the uncertainty analysis and the resultant clusters were called EDGE Zones. We then ran a General Linear Model to compare which variables significantly predicted the irreplaceability values within these EDGE Zone grid cells.

We explored the total number of species, EDGE species, threatened species, and endemic species (both grid cell endemics and zone-endemics) found in each EDGE Zone and compared the richness of these groups to a random selection of grid cells of a set size equal to the number of EDGE Zone grid cells (the random expectation). We then quantified the proportion of each species' total range that is contained within EDGE Zones and described the biome-type and number of intersecting ecoregions[86]. Using the global Multidimensional Poverty Index[53], we reported the deprivation faced by people within EDGE Zone countries. Using ANOVA with Tukey's Honest Significant Difference Test, we also tested which tetrapod groups have the greatest median ED scores, median EDGE scores, and the highest proportional contribution to the threatened evolutionary history in each grid cell relative to their richness, both within and outside of EDGE Zones.

To explore human pressure within each zone, we obtained data on the Human Footprint Index at a 1 km × 1 km resolution for both 1993 and 2009 from ref. 54. The Human Footprint Index is a spatial measure of human pressure on the land, factoring in eight different measures such as the presence of built environments, roads, and crop lands. The combined presence of these measures gives rise to a map of cumulative human pressure, with values ranging from 0 reflecting the lowest possible human pressure to a score of 50 reflecting the highest. We

categorised scores of below four as low human pressure and scores of above or equal to four as high human pressure as this threshold is indicative of when species are likely to become threatened by human land-use change[1,88–90]. For each EDGE Zone grid cell, we looked at the proportion of high human pressure scores. We then reported the proportion of each EDGE Zone that had shifted to high human pressure between measurements of the Human Footprint Index (1993 and 2009) and compared the proportion of high human pressure within EDGE Zones to the random expectation.

To explore the level of protection occurring within EDGE Zones, we downloaded data on the protected area (PA) network from the World Database on Protected Area[91]. Following WDPA guidelines, PAs whose status was proposed or not reported were removed and, for PAs where only point data was provided, we added a circular buffer zone around each point of a size equal to its reported area. The remaining sites were unionised to dissolve the boundaries between each polygon to prevent the double counting of overlapping areas. We then overlaid the resultant PA polygon with the rasterized grid used in the study and calculated the PA percentage cover of each EDGE Zone. To contrast how the protection status affects the overall coverage, we contrasted strict protection standards using PAs in IUCN categories I–IV (where 1a denotes a strict nature reserve and IV denotes a habitat/species management area) with relaxed standards by using all PAs in the dataset. We then compared the level of protection across all EDGE Zones to two thresholds using t-tests; 17%, reflecting target 11 of the Aichi Biodiversity Targets for 2020[55], and 30%, reflecting the Kunming-Montreal Global Biodiversity Framework target 3[56]. Finally, we assessed how the human pressure varies between protected and non-protected portions of EDGE Zones.

### Reporting summary

Further information on research design is available in the Nature Portfolio Reporting Summary linked to this article.

## Data availability

Species distribution data was obtained from the IUCN Red List of Threatened Species (Version 2021.1) for terrestrial and freshwater mammals and amphibians[79] (https://www.iucnredlist.org/resources/spatial-data-download), from BirdLife International (Version 2020.1) for birds[80] (https://datazone.birdlife.org/species/requestdis), and from the Global Assessment on Reptile Distributions (Version 1.5) for reptiles[29] (http://www.gardinitiative.org/data.html). We used publicly available EDGE scores from Gumbs et al.[45] supplementary data (https://figshare.com/s/09ab68484ba1e49cba48). The Global Multidimensional Poverty Index[53] was downloaded from: https://ophi.org.uk/. The Human Footprint Index[54] was downloaded from: https://doi.org/10.5061/dryad.052q5. Protected area[91] data was downloaded from: (https://www.protectedplanet.net/). Study outputs are available at: https://doi.org/10.6084/m9.figshare.25736703. All Supplementary Data 1–10 are available with this manuscript and on FigShare. Source data contains all the data needed for main and Supplementary Figs. Source data are provided with this paper.

## Code availability

The R code to run the EDGE Zone prioritisation procedure is available at https://doi.org/10.5281/zenodo.13254096.

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

## Acknowledgements

S.P. was funded by the NERC Science and Solutions for a Changing Planet Doctoral Training Programme (grant number NE/S007415/1), the CASE component of which was funded by On the Edge. We thank members of On the Edge, the Zoological Society of London's EDGE of Existence programme, and the IUCN SSC Phylogenetic Diversity Task Force for their constructive feedback and conceptual input in the making of this research.

## Author contributions

S.P., J.E.M.B., N.O. and R.G. conceived the study. S.P., N.O. and R.G. designed the analyses. S.P. and R.G. conducted the analyses. R.G. provided EDGE data. J.E.M.B., A.B., L.J.P., N.O. and R.G. provided technical support and conceptual advice. N.O. and R.G. supervised the study. S.P. wrote the paper, with substantial contributions from J.E.M.B., A.B., L.J.P., N.O. and R.G.

## Competing interests

The authors declare no competing interests.
