## [Peer Review File · Nature Communications]

Advancing EDGE Zones to identify spatial conservation priorities of tetrapod evolutionary historyReviewers' Comments:

Reviewer #1:

Remarks to the Author:

- Reproducibility

The reproducibility is not well guaranteed as the raw or processed data is rather difficult to obtain. The code on <https://github.com/rgumbs/EDGE2> presents the data flow, but the phylogenies are sourced from a preprint ([<https://doi.org/10.21203/rs.3.rs-2835356/v1>])(<https://doi.org/10.21203/rs.3.rs-2835356/v1>) that references the original sources. Since it is not a peer-reviewed paper, it would be better to present the topologies as additional material or explicitly cite the source for each taxonomic group.

- Areas Chosen

The use of political boundaries might not be the best approach at the global level. Instead, ecological or biogeographical boundaries should be considered (see Hughes et al. 2021). The areas chosen for the analyses are too broad and not entirely informative. An ecosystem-based approach, possibly using WWF categories or a grid level, would be more effective. This would allow for differentiation between lowlands and highlands, coastal and terra firme zones, etc. The authors used grids but did not specify the grid size and preferred a nationwide approach without further explanation.

- Richness

PD and EDGE have shown correlation with species richness. AvTD could serve as an auxiliary tool to assess the bias given the richness and the impact of not including the IUCN categories. This helps evaluate whether the authors are assessing richness, but using EDGE2 as a surrogate.

- IUCN Validity

My concern with EDGE in general is that the IUCN is not comprehensively well-curated for all groups and is outdated. Although the Red List is the best available tool for assessing the extinction risk of species, it suffers from being quite slow in the assessment process and lacks coverage for numerous species. The list is biased towards certain species and does not adequately reflect the threats facing many species. As Borgelt et al suggest: "more than half of the data-deficient species are predicted to be threatened by extinction" (see Borgelt et al. 2022, Palacio et al. 2023).

- References

Borgelt, J., Dorber, M., Høiberg, M.A. et al. More than half of data deficient species predicted to be threatened by extinction. *Commun Biol* 5, 679 (2022). <https://doi.org/10.1038/s42003-022-03638-9>

Palacio et al (2023) [DOI: 10.22541/au.169945445.50394320/v1](https://d197for5662m48.cloudfront.net/documents/publicationstatus/177025/preprint_pdf/2af44df99ed5dd465704a61adc8cdf2b.pdf)

Reviewer #2:

Remarks to the Author:

Dear Editor

Please find below my review of manuscript #NCOMMS-23-53254 “Advising EDGE Zones to identify spatial conservation priorities of tetrapod evolutionary history”. This study provides updated patterns of EDGE Zones for tetrapod. The results identify 25 tetrapod EDGE Zones, all located in terrestrial environments. The manuscript is well written, the analysis are exhaustive and sound, and the overall result provide an important criterion for managers and others involved in conservation and management. I congratulate the authors for this important contribution. I only have minimal comments and recommend acceptance with minor changes.

Abstract

This result “We find that 32 threatened evolutionary history peaks in Cameroon, whilst EDGE species richness peaks in 33 Madagascar” is important but given all the analysis around EDGE zones I think this sentence can be remove and highlight more about findings related to the 25 tetrapod EDGE Zones such as that these zones are located over 117 ecoregions and 10 different biomes.

Introduction

Line 90. Add mapping of the ‘spatial’ distributions of EDGE species.

Results

I recommend moving supplementary Table 1 as part of the main manuscript for readers to directly see what the 25 EDGE Zones in addition to their spatial location (Fig.3) are.

Discussion

Line 212. How many EDGES Zones overlap with the 29 countries from Multidimensional Poverty Index data?

Final thoughts

I wonder if the top 25 EDGE zones would change when considering all jawed vertebrates (Gumbs et al., 2023), especially given that sharks are by far the most evolutionary distinct of all major

radiations of jawed vertebrates (see Stein et al. 2018 paper)

<https://www.nature.com/articles/s41559-017-0448-4>. Given the biodiversity crisis we want to protect EDGE zones that maximize the protection of EDGE species, and in the future, a global analysis including bony and cartilaginous fish would be important to determine if there are EDGE zones in the marine realm. I recommend the authors to consider adding a sentence on this regard to their conclusion.

Reviewer #3:

Remarks to the Author:

I have revised the manuscript entitled “Advancing EDGE Zones to identify spatial conservation priorities of tetrapod evolutionary history”. This is a very interesting approach to implement practical conservation efforts based on averting the loss of evolutionary history. The research group presenting this has been working for a long time on studying and promoting the use of EDGE and has an active participation on conservation politics as well.

The manuscript is very well-written, and methods are well explained. However, there are sections that I find a bit repetitive. Maybe the authors could try to summarize some sections that seem to be explained more than once, especially in the results and methods sections. Another area where authors could condense the text is lines 52-58 in the introduction, when talking about benefits and option values.

I have a few other comments and suggestions, that hopefully will help improve the manuscript.

The first doubt that arises when reading this, is the relationship with the recently published EDGE2 index. The fact that calculations were made with EDGE2 are not clear until well into the manuscript. I think a decision needs to be made here in terms of communication, because EDGE and EDGE2 are calculated differently, and in Gumbs et. al. (2023) regarding EDGE2, a strong argument is made as to why we should in fact use the new index (which I agree with). It would be good for the users of the index if a difference is made between the two when mentioning them. In the methods it is argued that the EDGE Zones approach should not be difficult to implement given the availability of EDGE scores (lines 521-522), however these scores are EDGE and not EDGE2, which leads to confusion. This should be explained at the beginning of this manuscript or just treat the index as EDGE2 for clarity.

Line 62 in the introduction. I do not believe that PD can determine variability within species, or at least it has not been extensively used for this. If you still want to make that argument, I suggest adding an explanation.

Line 69, there is an extra parenthesis that needs to be eliminated.

Line 204 and 236. It would be interesting perhaps to compare these areas just with areas of high tetrapod endemism (the traditional measure), or at least discuss how that comparison would be

like. For people that believe evolutionary measures are overly complicated, an empirical argument needs to be made as to why they provide more information than traditional measures.

Line 491. It says here that EDGE2 scores used were generated in Gumbs et al 2023, however that paper only mentions EDGE2 scores for mammals. Where the other tetrapod scores obtained de novo?

Line 518. I have a methodological doubt here. As far as I understand, summing up EDGE scores, means that you are counting branches more than once in any given area. This is probably not a problem for distantly related taxa, but it might overestimate the extinction risk when a given grid cell harbors taxa that are more closely related.

Dear anonymous reviewers,

We thank you for your review of our manuscript. We are appreciative of the positive affirmation received on its quality and for the valuable suggestions and important concerns raised. We have addressed your comments with a point-by-point analysis, given below.

In particular, we have made the origin of our data more explicit by citing the source phylogenies underpinning our data and providing the link to the figshare repository housing the EDGE scores used. Since the submission of our paper, the article that the EDGE scores were taken from has been published in Nature Communications (<https://doi.org/10.1038/s41467-024-45119-z>) and we have updated our manuscript in light of this. Furthermore, we have added an additional analysis highlighting the EDGE species richness found within ecoregions and have made it clearer that EDGE data reflects the probable elevated extinction risks of Data Deficient species, mitigating against being over-reliant on IUCN Red List data. We hope that these have addressed reviewer #1's technical concerns. Additionally, we have made a specific reference to the potential expansion of the EDGE concept to include different taxonomic groups and ecological realms, as per reviewer #2's suggestion. Finally, in response to reviewer #3, we have removed the unnecessary repetition in our methods, added an analysis comparing our approach to one based on Weighted Endemism, and clarified the methodological concern of summing EDGE scores. We have made all data underpinning our results and output available in a figshare repository through the ncomms submission portal, the link to which will be made available on re-submission.

We thank you again for their time and expertise. We believe your constructive feedback has significantly improved the rigor and overall quality of our paper.

Best wishes

Sebastian

REVIEWER COMMENTS

Reviewer #1 (Remarks to the Author):

- Reproducibility

The reproducibility is not well guaranteed as the raw or processed data is rather difficult to obtain. The code on <https://github.com/rqumbs/EDGE2> presents the data flow, but the phylogenies are sourced from a preprint ([<https://doi.org/10.21203/rs.3.rs-2835356/v1>])(<https://doi.org/10.21203/rs.3.rs-2835356/v1>) that references the original sources. Since it is not a peer-reviewed paper, it would be better to present the topologies as additional material or explicitly cite the source for each taxonomic group.

We agree and support all data being open. The research article containing the methods, sources and results for all ED and EDGE scores used here has now been published (<https://doi.org/10.1038/s41467-024-45119-z>). The EDGE scores from that paper, used here, are freely available from: <https://figshare.com/s/09ab68484ba1e49cba48>. We have also now directly cited the original phylogenies that the EDGE2 scores were derived from in the text on lines [555-560] to ensure the reproducibility of our results.

- Areas Chosen

The use of political boundaries might not be the best approach at the global level. Instead, ecological or biogeographical boundaries should be considered (see Hughes et al. 2021). The areas chosen for the analyses are too broad and not entirely informative. An ecosystem-based approach, possibly using WWF categories or a grid level, would be more effective. This would allow for differentiation between lowlands and highlands, coastal and terra firme zones, etc. The authors used grids but did not specify the grid size and preferred a nationwide approach without further explanation.

We agree that national level alone would not be sufficient for this work, and this was something we only did for our policy-relevant analysis of the number of EDGE species in each country. All other analyses of evolutionary history were calculated at the grid cell level with a resolution of 96.5 km x 96.5 km, which was first stated on line 131, where we say:

“Maximum EDGE species richness occurs in Northern Madagascar, with a single 96.5 km x 96.5 km grid cell containing 45 EDGE species, but EDGE species are absent from parts of Central Australia, Central Europe, and the Sahara Desert.”

For EDGE species richness, we aggregated to the national level to align with global policy and action. For example, the EDGE index (<https://doi.org/10.1111/cobi.14138>) in the Kunming-Montreal Global Biodiversity Framework is designed to be reported at the national and global level.

We agree with the suggestion that an ecosystem-based approach can add depth to a global mapping study. In our study, we had initially intersected EDGE Zones with ecoregions [229] to highlight the breadth of ecosystems covered. We have now added an additional analysis (Supplementary Fig. 1) detailing the EDGE species richness found within each ecoregion, which we have reported in the results [125:127]:

“Their [EDGE Species] collective distribution covers 92.9% of the world’s terrestrial surface (Fig. 1a) and 833 ecoregions (median: 15 species per ecoregion; IQR: 8-27 species; Supplementary Fig. 1).”

And in the methods, where we also made the identity of the EDGE species found within each ecoregion available in a figshare repository [570-574]:

“We then intersected EDGE species distributions with ecoregions, finding that there were 33 EDGE species that did not overlap with any ecoregion; these small-island endemic EDGE species had distributions that were too remote to be covered by the coarse designation of ecoregions. We have made the list of EDGE species found within each ecoregion available (see ‘Data availability’).”

- Richness

PD and EDGE have shown correlation with species richness. AvTD could serve as an auxiliary tool to assess the bias given the richness and the impact of not including the IUCN categories. This helps evaluate whether the authors are assessing richness, but using EDGE2 as a surrogate.

We acknowledge that species richness, PD, and EDGE are highly correlated at a global level, but our existing analyses, which were designed to determine exactly whether EDGE simply reproduced species richness patterns, indicate that this correlation is weakened when we focus in on priority grid cells. We discussed and deconstructed this in the results, where we say:

“The distribution of threatened evolutionary history was highly correlated with species richness (Pearson’s correlation: $r = 0.916$, e.d.f. = 36.3, $p < 0.0001$). However, priority regions resulting from the two measures are more dissimilar than the high correlation suggests, with the congruence decreasing at high percentiles. For example, for the 80th percentile (i.e., the 20% of highest scoring grid cells), the two have an 84.8% overlap. However, at the 90th percentile this decreases to an overlap of 69.8%, followed by 49.4% at the 97.5th percentile. Areas of high species richness dominate large parts of the Amazon basin and the Atlantic Forest, reflecting widely distributed shared species compositions (Supplementary Fig. 5). Meanwhile, areas of high threatened evolutionary history at the 97.5th percentile show a wider array of geographic locations, including Southeast Asia, Cameroon, Madagascar, the Eastern Arc, the Western Ghats, and Sri Lanka (Supplementary Fig. 5). This dissimilarity in high scoring areas underlines the importance of considering multiple facets of biodiversity within conservation prioritisations^{30,50,51}.” [186-197]

We are also explicit that our goal of prioritising grid cells included the rationale of capturing the most threatened evolutionary history whether driven by large numbers of species in general or driven by a small number of high-scoring EDGE species:

“Our method prioritised for large accumulations of unique threatened evolutionary history, whether driven by a small number of highly distinct species or a larger number of less distinctive but more threatened species as both represent important potential losses from the Tree of Life.”

However, we note that this originally appeared too late in the manuscript and have therefore moved and the line from the methods into the Introduction [104-107] to make our rationale clearer from the outset.

- IUCN Validity

My concern with EDGE in general is that the IUCN is not comprehensively well-curated for all groups and is outdated. Although the Red List is the best available tool for assessing the extinction risk of species, it suffers

from being quite slow in the assessment process and lacks coverage for numerous species. The list is biased towards certain species and does not adequately reflect the threats facing many species. As Borgelt et al suggest: "more than half of the data-deficient species are predicted to be threatened by extinction" (see Borgelt et al. 2022, Palacio et al. 2023).

- References

Borgelt, J., Dorber, M., Høiberg, M.A. et al. More than half of data deficient species predicted to be threatened by extinction. *Commun Biol* 5, 679 (2022). <https://doi.org/10.1038/s42003-022-03638-9>

Palacio et al (2023) [DOI: 10.22541/au.169945445.50394320/v1](https://d197for5662m48.cloudfront.net/documents/publicationstatus/177025/preprint_pdf/2af44df99ed5dd465704a61adc8cdf2b.pdf)

As much as we sympathise with the reviewer's perspective on the IUCN Red List, we also agree that it is the best available tool for assessing extinction risk. It also currently underpins the extinction risk component of EDGE2 scores. We have now added a line to the discussion around the benefits of using alternative conservation tools to determine extinction risk, where we now mention the recent advances of using machine learning extinction risk predictions for seed plants:

"Furthermore, in these groups, tools such as machine learning algorithms are allowing for comprehensive extinction risk predictions, helping to mitigate against the IUCN Red List's limited coverage." [lines 525-527]

We have also made it clearer that Data Deficient and unevaluated species are incorporated by EDGE2 and are assigned estimated probabilities of extinction analogous to a threatened categorization, which we have now included in our introduction:

"The EDGE2 approach also allow for the incorporation of species with inadequate data: those lacking phylogenetic information are imputed across a distribution of trees, and those missing extinction risk data receive estimated probabilities of extinction with a median comparable to the Vulnerable IUCN categorisation." [lines 89-92]

We believe that this threatened designation of unevaluated species, quoted above, helps mitigate against the overreliance on Red List assessments. Furthermore, our study incorporates 92.4% of described tetrapod animals [line 157], which we believe makes the resultant patterns broadly reliable.

Reviewer #2 (Remarks to the Author):

Dear Editor

Please find below my review of manuscript #NCOMMS-23-53254 "Advising EDGE Zones to identify spatial conservation priorities of tetrapod evolutionary history". This study provides updated patterns of EDGE Zones for tetrapod. The results identify 25 tetrapod EDGE Zones, all located in terrestrial environments. The manuscript is well written, the analysis are exhaustive and sound, and the overall result provide an important criterion for managers and others involved in conservation and management. I congratulate the authors for this important contribution. I only have minimal comments and recommend acceptance with minor changes.

We thank the reviewer for their positive thoughts on our work.

Abstract

This result "We find that 32 threatened evolutionary history peaks in Cameroon, whilst EDGE species richness peaks in 33 Madagascar" is important but given all the analysis around EDGE zones I think this sentence can be remove and highlight more about findings related to the 25 tetrapod EDGE Zones such as that these zones are located over 117 ecoregions and 10 different biomes.

We appreciate the reviewer's suggestion to focus more on EDGE Zones in the abstract and have made the recommended change. [35-36]

Introduction

Line 90. Add mapping of the 'spatial' distributions of EDGE species.

We have made the recommended insertion.

Results

I recommend moving supplementary Table 1 as part of the main manuscript for readers to directly see what the 25 EDGE Zones in addition to their spatial location (Fig.3) are.

Whilst we agree with the reviewer that it is useful to contextualise the spatial locations of EDGE Zones, we believe that moving Table S1 into the main manuscript would lead to the repetition of information already contained within Fig 4. Specifically, Table S1 contains information on EDGE species richness, endemic species richness, species richness, threatened evolutionary history, human pressure, and protection levels – all besides species richness are presented within Figure 4. Given we are already at the limit of figures/tables allowed in the main text, to bring in the table would mean to remove figure 4.

Discussion

Line 212. How many EDGES Zones overlap with the 29 countries from Multidimensional Poverty Index data?

29 countries spread across 24 EDGE Zones, which we have now added into the manuscript [232:234]:

"Of the 109 developing countries with global Multidimensional Poverty Index data⁵², which assess a person's combined deprivation in health, living standards, and education, 29 of these overlap with 24 EDGE Zones (all except the Australian EDGE Zone)."

Final thoughts

I wonder if the top 25 EDGE zones would change when considering all jawed vertebrates (Gumbs et al., 2023), especially given that sharks are by far the most evolutionary distinct of all major radiations of jawed vertebrates (see Stein et al. 2018 paper) <https://www.nature.com/articles/s41559-017-0448-4>. Given the biodiversity crisis we want to protect EDGE zones that maximize the protection of EDGE species, and in the future, a global analysis including bony and cartilaginous fish would be important to determine if there are EDGE zones in the marine realm. I recommend the authors to consider adding a sentence on this regard to their conclusion.

There are plans in the pipeline to expand the EDGE Zone concept to the marine world, incorporating both ray-finned fish and sharks, along with further plans to consider freshwater ecosystems and plant groups. To enable this expansion, we have now specifically referred to the EDGE Zones highlighted by our study as Tetrapod EDGE Zones, allowing for further inclusions down the line. Whilst these expansions were not within the scope of this study, we agree that this is an important point to mention and have added the following line at the end of the discussion [522-525]:

"Third, our method of summing EDGE scores offers the potential for extending this approach to other important taxonomic groups, such as seed plants and fish 39,45,76,77, and therefore allows for the expansion of EDGE Zones into freshwater and marine realms."

Reviewer #3 (Remarks to the Author):

I have revised the manuscript entitled “Advancing EDGE Zones to identify spatial conservation priorities of tetrapod evolutionary history”. This is a very interesting approach to implement practical conservation efforts based on averting the loss of evolutionary history. The research group presenting this has been working for a long time on studying and promoting the use of EDGE and has an active participation on conservation politics as well.

The manuscript is very well-written, and methods are well explained. However, there are sections that I find a bit repetitive. Maybe the authors could try to summarize some sections that seem to be explained more than once, especially in the results and methods sections. Another area where authors could condense the text is lines 52-58 in the introduction, when talking about benefits and option values.

We thank the reviewer for their positive and constructive feedback on the manuscript. On reflection, we agree that there are areas of reiteration within our manuscript, particularly in our methods and results sections. We have therefore made the following changes:

- We have deleted a line in the methods [592-595] that repeated the precedent of using species specific scores for mapping evolutionary history, which was already mentioned in the results [158-161]. This line also repeated the advantage of using publicly available EDGE2 scores, which was mentioned in the discussion [405-406]*
- We trimmed our introduction of spatial complementarity in the results [201-204], which are described in more detail in the methods [614 onwards].*
- We shortened our introduction of irreplaceability in the results [208-211] which is then described more fully in the methods [647-650]*
- We trimmed our explanation of PD benefits and option value in the introduction [52-61.]*

I have a few other comments and suggestions, that hopefully will help improve the manuscript.

The first doubt that arises when reading this, is the relationship with the recently published EDGE2 index. The fact that calculations were made with EDGE2 are not clear until well into the manuscript. I think a decision needs to be made here in terms of communication, because EDGE and EDGE2 are calculated differently, and in Gumbs et al. (2023) regarding EDGE2, a strong argument is made as to why we should in fact use the new index (which I agree with). It would be good for the users of the index if a difference is made between the two when mentioning them. In the methods it is argued that the EDGE Zones approach should not be difficult to implement given the availability of EDGE scores (lines 521-522), however these scores are EDGE and not EDGE2, which leads to confusion. This should be explained at the beginning of this manuscript of just treat the index as EDGE2 for clarity.

We agree that this needs clarification. Our study made use of EDGE2 scores (Gumbs et al. 2023a; <https://doi.org/10.1371/journal.pbio.3001991>) and referred to them as ‘EDGE scores’ to be in line with the recently published paper on jawed vertebrates in Nat comms, which produced the data underpinning this study and referred to the scores as ‘EDGE scores under the EDGE2 protocol’ (<https://doi.org/10.1038/s41467-024-45119-z>). This is in line with other recent papers that have used the phrasing of ‘EDGE scores under the EDGE2 protocol’, such as McClure et al. 2023 (<https://doi.org/10.1111/cobi.14141>) and Gumbs et al. 2023b (<https://doi.org/10.1111/cobi.14138>).

To make this clearer, we have now mentioned in the introduction that we used EDGE2 scores but refer to them as EDGE scores [83-89]:

“The approach has recently been updated under the EDGE2 methodology⁴¹ to incorporate phylogenetic complementarity⁴¹, which describes how the irreplaceability of focal species is influenced by the extinction risk of closely-related species^{42,43}. Species with many secure close relatives are considered less irreplaceable than those with few and highly threatened relatives. As such, EDGE2 scores (hereafter, referred to as EDGE scores) now quantify threatened evolutionary history, with the scores representing the amount of PD expected to be lost, in millions of years (MY), that can be averted with conservation action⁴¹.”

Line 62 in the introduction. I do not believe that PD can determine variability within species, or at least it has not

been extensively used for this. If you still want to make that argument, I suggest adding an explanation.

An error on our part - we agree that PD calculations do not inform on the variability within a species and that our phrasing was incorrect. We meant to write "of and between", referring to the terminal branch lengths unique to a species and the branch lengths shared between them. On consideration, we have changed the line to the more explicit "unique and shared evolutionary history of species" [65].

Line 69, there is an extra parenthesis that needs to be eliminated.

Done!

Line 204 and 236. It would be interesting perhaps to compare these areas just with areas of high tetrapod endemism (the traditional measure), or at least discuss how that comparison would be like. For people that believe evolutionary measures are overly complicated, an empirical argument needs to be made as to why they provide more information than traditional measures.

We have added an additional analysis [Supplementary Fig. 11] that speaks to your point, brought up in the results [251-255], methods [640-645] and in the discussion [433--447] as:

"Using complementarity also ensured that the EDGE Zones captured disparate species compositions but unlike an approach that selects just for weighted endemism, EDGE Zones efficiently prioritize areas for their unique evolutionary history as well; we found that only 56.3% of grid cells were shared between our chosen approach and one that focussed on weighted endemism only (Supplementary Fig. 11), with the latter also requiring 15.6% more grid cells to capture 6.8% less threatened evolutionary history. We found that this endemism prioritisation over-selected across the Andes, which displays high species turnover but shows diminishing returns on the amount of new phylogenetic diversity represented. Furthermore, it neglected areas of high richness and low grid-cell endemism (e.g. the Amazon) and areas of low richness and high phylogenetic diversity (e.g. Seychelles). We believe this underpins the strength of prioritising using summed EDGE scores, which highlights areas of large concentrations of unique and threatened evolutionary history, whether primarily driven by threat, richness, endemism, or evolutionary distinctiveness."

Line 491. It says here that EDGE2 scores used were generated in Gumbs et al 2023, however that paper only mentions EDGE2 scores for mammals. Where the other tetrapod scores obtained de novo?

EDGE2 scores were taken from a recently published manuscript on jawed vertebrates (Gumbs et al. 2024; <https://doi.org/10.1038/s41467-024-45119-z>). We have made this more explicit and referenced the source phylogenies for greater clarity on the data used [555-560]:

"We used EDGE scores from Gumbs et al.⁴⁵, generated using the updated EDGE2 approach⁴¹, which sourced phylogenetic data from Jetz and Pyron for amphibians⁸⁰, Jetz et al. for birds³⁶, Upham et al. for mammals⁸¹, Tonini et al. for squamates⁸², and Colston et al. for crocodiles and turtles⁸³."

Line 518. I have a methodological doubt here. As far as I understand, summing up EDGE scores, means that you are counting branches more than once in any given area. This is probably not a problem for distantly related taxa, but it might overestimate the extinction risk when a given grid cell harbors taxa that are more closely related.

This over-counting of branch lengths when using summed scores is something we considered carefully when designing our study. We have now added the following to the discussion addressing this point [393-412]:

"We elected to use summed EDGE scores over a branch-based approach to calculate threatened evolutionary history. The two approaches were highly correlated and displayed 98% overlap when selecting EDGE Zone priority grid cells. Summing EDGE scores does mean there will be instances of overcounted branch lengths. However, due to the way EDGE scores are calculated, this double counting is most acute for Critically Endangered sister species and becomes increasingly inconsequential as related species become less threatened. For example, an internal branch subtended by at least one Least Concern species can only contribute a maximum 6% of its total length⁴¹, as the low extinction risk of the Least Concern species effectively safeguards its descendant branches against extinction. Summing EDGE scores can therefore slightly

overestimate the amount of expected loss in areas with large concentrations of highly threatened, closely related species. Given the large potential losses to deep evolutionary history that this could entail^{63,64}, we are not concerned by this methodological artefact. Furthermore, we believe the summed EDGE2 approach provides a clear benefit to policy makers and conservation practitioners. This approach, which uses publicly available data⁴⁵, is more accessible to non-specialists both in theory and application and can be computed at a fraction of the time of branch-length-dependent methods, providing a feasible avenue to incorporate evolutionary history more readily within conservation planning and decision making. Considering species are the typical unit of conservation, using species-specific scores also allows for the inclusion of other species-level biodiversity measures, such as cultural value and functional traits^{52,65}, or the incorporation of species from disparate clades into prioritisation exercises³⁶.”

Furthermore, we mention the 98% overlap between the branch-length approach and the summed-score approach for selecting EDGE Zone grid cells in the results [247-250] and display in the SM [Supplementary Fig. 9]:

“There was a 97.7% overlap between EDGE Zone priority cells and cells selected using phylogenetic branch-length calculations of expected PD loss (Supplementary Fig. 9), with the two methods showing a strong correlation in the frequency in which cells were selected ($\rho = 0.958$, $p < 0.0001$)” [243-246]

And the strong correlation between the two approaches is mentioned here [161-166]:“

We found an extremely strong positive correlation between both the phylogenetic branch length approach and the summed EDGE2 score approach for calculating threatened evolutionary history for each tetrapod group (Pearson’s correlation adjusted for spatial autocorrelation^{29,47-49}; all $p > 0.99$) (Supplementary Text 1), as well as a high overlap in the location of priority grid cells at the 90th (>94.8% overlap) and 95th percentile (>94.4% overlap) for each taxonomic group (Supplementary Text 1).”

Reviewers' Comments:

Reviewer #1:

Remarks to the Author:

The authors have addressed all the concerns I raised about methods and potential bias. In its current state, this is a good paper, and I recommend its publication after a sample data set is included in the GitHub site to test the code.

Reviewer #2:

Remarks to the Author:

Dear Authors

I have reviewed the revised manuscript and I have not further edits or recommendations. As I stated in my initial review. The analysis in this study are exhaustive and sound. Furthermore, the identification of Tetrapod EDGE Zones by country provides an important criterion for local managers and governments in the conservation and management of tetrapods.

Reviewer #3:

Remarks to the Author:

I have revised again the second version of the manuscript "Advancing EDGE Zones to identify spatial conservation priorities of tetrapod evolutionary history". I thank the authors for addressing all my concerns and suggestions, as well as the others presented by the rest of the reviewers. I believe the manuscript has substantially improved and I have no further concerns.

Dear reviewers,

Thank you for your continued feedback throughout the revision process. We believe our manuscript has been greatly strengthened through your help and we are grateful for the time you gave up to this end.

We agree with reviewer 1 on the need for a sample dataset to accompany our code. We have therefore provided all required data to run the EDGE Zone prioritisation for mammals. This data is accessible through the Figshare repository, linked in our code and manuscript. As the figshare repository only goes live with publication, the dataset can be accessed before here.

Thank you once again for your valuable feedback and the time you have invested in reviewing our manuscript. Your insights have been very helpful in refining our work and we greatly appreciate your contribution to this process.

Best wishes

Sebastian

REVIEWERS' COMMENTS

Reviewer #1 (Remarks to the Author):

The authors have addressed all the concerns I raised about methods and potential bias. In its current state, this is a good paper, and I recommend its publication after a sample data set is included in the GitHub site to test the code.

Reviewer #1 (Remarks on code availability):

I reviewed the code for consistency in naming conventions, commenting style, and overall structure. The code appears well-organized at this level. However, I was unable to run the code to test functionality because the required input data is not currently available on the GitHub repository. If possible, including a sample input data file would allow for a more thorough review.

This is an important suggestion and we have now included a sample dataset. Thank you!

Reviewer #2 (Remarks to the Author):

Dear Authors

I have reviewed the revised manuscript and I have not further edits or recommendations. As I stated in my initial review. The analysis in this study are exhaustive and sound. Furthermore, the identification of Tetrapod EDGE Zones by country provides an important criterion for local managers and governments in the conservation and management of tetrapods.

Thank you!

Reviewer #3 (Remarks to the Author):

I have revised again the second version of the manuscript "Advancing EDGE Zones to identify spatial conservation priorities of tetrapod evolutionary history". I thank the authors for addressing all my concerns and suggestions, as well as the others presented by the rest of the reviewers. I believe the manuscript has substantially improved and I have no further concerns.

Thank you!